# Molecular Dynamics of Lysine Dendrigrafts in Methanol–Water Mixtures

**DOI:** 10.3390/ijms24043063

**Published:** 2023-02-04

**Authors:** Emil I. Fatullaev, Oleg V. Shavykin, Igor M. Neelov

**Affiliations:** 1School of Computer Technologies and Control, St. Petersburg National Research University of Information Technologies, Mechanics and Optics (ITMO University), Kronverkskiy pr. 49, 197101 St. Petersburg, Russia; 2Physics Department, Lomonosov Moscow State University, Leninskie Gory 1-2, 119991 Moscow, Russia; 3Department of Mathematics, Tver State University, Sadoviy per. 35, 170102 Tver, Russia; 4Institute of Macromolecular Compounds of the Russian Academy of Sciences, 199004 St. Petersburg, Russia

**Keywords:** lysine dendrigraft, computer simulation, molecular dynamics, secondary structure, methanol/water mixtures, hydrogen bonds

## Abstract

The molecular dynamics method was used to study the structure and properties of dendrigrafts of the first and second generations in methanol–water mixtures with various volume fractions of methanol. At a small volume fraction of methanol, the size and other properties of both dendrigrafts are very similar to those in pure water. A decrease in the dielectric constant of the mixed solvent with an increase in the methanol fraction leads to the penetration of counterions into the dendrigrafts and a reduction of the effective charge. This leads to a gradual collapse of dendrigrafts: a decrease in their size, and an increase in the internal density and the number of intramolecular hydrogen bonds inside them. At the same time, the number of solvent molecules inside the dendrigraft and the number of hydrogen bonds between the dendrigraft and the solvent decrease. At small fractions of methanol in the mixture, the dominant secondary structure in both dendrigrafts is an elongated polyproline II (PPII) helix. At intermediate volume fractions of methanol, the proportion of the PPII helix decreases, while the proportion of another elongated β-sheet secondary structure gradually increases. However, at a high fraction of methanol, the proportion of compact α-helix conformations begins to increase, while the proportion of both elongated conformations decreases.

## 1. Introduction

Ideal dendrimers are regular, spherical, hyperbranched molecules emanating from a single center and consisting of branched fork-shaped repeating units with one input handle and two (or less often) three output prongs. For this reason, the number of elements in each next generation doubles (or triples) and dendrimers have a large number of terminal groups available for functionalization. The first dendrimers were synthesized more than 40 years ago [1,2,3,4,5]. The unique structure of dendrimers has made them very popular for various applications [6,7,8,9,10,11,12,13,14,15]. Lysine dendrimers [16] composed of lysine amino acid residues are biocompatible and, therefore, well-suited for use in biomedicine. The properties of lysine dendrimers as well as their applications have been studied in many papers [17,18,19,20,21,22,23,24]. More complex lysine-based and general peptide dendrimers could include any amino acid residues and could be used in biomedical applications including drug and gene delivery [25,26,27,28,29,30,31,32,33,34,35,36,37,38,39,40,41,42,43,44,45,46,47,48,49].

At the same time, the synthesis of dendrimers, in general, and lysine dendrimers, in particular, involves many protection and deprotection steps, which makes it rather slow and expensive. That is why, in the early 2000s, the new methods for the synthesis of slightly less regular lysine dendrigraft molecules were elaborated [50]. In these new molecules, the core of is not a single lysine monomer, as in a lysine dendrimer, but a linear lysine chain consisting of eight lysine amino acid residues. This core is a lysine dendrigraft of the first generation (DGL1). The second-generation lysine dendrigraft (DGL2) consists of 48 lysines including DGL1 and linear lysine side chains attached to all terminal groups of DGL1 containing nitrogen. The lysine dendrigraft of generation 3–5 consists of DGL2 and dendritic side chains attached to its terminal groups containing nitrogen. They consist of an average of 123, 365, and 963 lysine monomers, respectively. Synthesis of dendrigrafts is described in [50,51,52], characterization of dendrigrafts, their experimental studies in [53,54,55,56,57,58,59,60,61,62,63,64], MD simulation in [65,66], drug and gene delivery [67,68,69,70,71,72,73,74,75,76,77,78,79,80,81,82,83,84,85,86], as well as other applications, including magnetic resonance imaging, catalysis, virus concentration in water, heparin antidote, surface coating, and dental materials in [87,88,89,90,91,92,93,94]. In the literature, there are many papers about the structure and properties of long linear lysine peptides in various solvents, including water and methanol as well as in their mixtures. It was shown that the sizes and secondary structures of polylysine macromolecules change significantly with the temperature and concentration of methanol [95,96]. However, it is also well known that the properties of polylysine peptides depend on their degree of polymerization [97]. Because DGL1 and DGL2 contain rather short lysine chains and DGL2 is a highly-branched molecule, their behaviors could significantly differ from those of long linear polylysine.

Lysine dendrigrafts in mixed solvents are also of interest for creating new amphiphilic block copolymers similar to amphiphilic linear–dendron block copolymers and dendron–dendron block copolymers (Janus dendrimers). In recent years, we have simulated both lysine/peptide dendrimers and lysine dendrigrafts, as well as lysine hybrid linear-dendron molecules [46], and molecules consisting of a lysine dendrimer with many hydrophobic tails [98]. The hybrid molecule used in the last article has a new topological structure of the lysine dendrimer–hydrophobic dendrigraft type. Molecules of the last two types have great prospects for use as delivery vehicles for drugs and genetic material. Micellization of such amphiphilic hybrid molecules of different topologies in different solvents, including mixed solvents (for example, methanol–water), occurs in different ways, so the study of these processes is of great theoretical and practical importance. This manuscript studies the lysine dendrigraft, which is a hydrophilic (positively charged) block of an amphiphilic hybrid molecules of new types (linear-dendrigraft or dendrigraft-dendrigraft types). The results obtained in this article also make it possible to understand whether the results obtained for linear polylysine molecules on conformational transitions under changing external conditions (including changes in the solvent composition) also apply to lysine dendrigrafts and whether these results should be taken into account when designing new drug carriers and genetic material based on lysine dendrigrafts and micelles from hybrid molecules containing dendrigraft blocks.

There are very few publications in the literature devoted to modeling dendrigrafts in water, and there are practically no publications describing their behaviors in other solvents beside DMF [53,57] and their mixtures with water, including mixtures with methanol. The goal of the present paper is to understand how the properties of the dendrigrafts of first- and second-generations change with the change in volume fractions of methanol in methanol–water mixtures.

## 2. Results and Discussion

Snapshots of DGL1 and DGL2 dendrigrafts at the end of the molecular dynamic simulation of DGL1 (Figure 1) and DGL2 (Figure 2) at each fraction ϕ of methanol were prepared using the molecular editor PyMOL. A visual comparison of these images shows a weak dependence of sizes on ϕ in the range ϕ = 0.2–0.6 and a noticeable decrease in size at ϕ = 0.7 and higher. The asphericity parameter α [40,41,99,100,101] used to describe the shape of dendrigrafts (see Table 1) was very small for both DGLs at all fractions of methanol. It means that the shape of both DGLs is close to spherical one.

We used several options for estimating the size of the dendrigraft. The first option is the mean-squared gyration radius $R_{g}$, which is calculated as:(1)$$R_{g}={\left(\frac{1}{M}\sum_{i}m_{i}r_{i}^{2}\right)}^{1/2},$$
where $M,\;m_{i}$ are the molecular masses of the dendrigraft and the *i*-th atom, correspondingly, and $r_{i}$ is the distance from the *i*-th atom to the center of mass of the dendrigraft. Figure 3a demonstrates the dependence of the gyration radius $R_{g}$ on the methanol fraction ϕ in the solution. It can be seen that the gyration radius $R_{g}$ for both dendrigrafts practically does not change in the intervals of 0.2–0.6 and decreases with an increase in the methanol fraction ϕ≥ 0.7 in agreement with behaviors of DGL1 in Figure 1 and DGL2 in Figure 2.

Unlike the dendrimer, which has a single central core, the dendrigraft has a linear core consisting of eight lysine amino acid residues. Therefore, the size of the lysine dendrigraft can vary much more strongly depending on the core conformation than the size of the dendrimer. It is known that the size of a second-generation dendrigraft in water is approximately 2.2 times larger than in dimethylformamide [57]. Thus, the use of a dendrigraft instead of a dendrimer for drug delivery, in addition to the cheapness of a dendrigraft, provides an additional possibility of a strong directed change in its size, and internal structure by changing the composition of the solvent. A change in the size of a lysine dendrigraft, as well as a change in the size of the linear polylysine chain, can occur by changing the secondary structures of the linear peptide chains that make up the dendrigrafts. Since the dielectric permittivity of methanol (32.7) and water (80) differs by almost 2.5 times, the dielectric permittivity of the mixture will decrease from 80 to 32.7 when the fraction of methanol increases from 0 to 100% and becomes close to the value at which the formation of the ion pair begins in the system. The Bjerrum length, which determines the system parameters at which counterions condense on a polyelectrolyte molecule (in our case, a dendrigraft) in a given solvent, is inversely proportional to the dielectric permittivity of the solvent. Therefore, when the proportion of methanol in the solution changes from 0 to 100%, this value will increase from about 7 to about 17 nm. The decrease in the dendrigraft size ($R_{g}$) begins at ϕ=0.7 for DGL2 (see Figure 3a), while for DGL1 it is less pronounced and starts at ϕ=0.9. Thus, partial condensation of counterions on the positively charged NH${}_{3}^{+}$ groups of DGLs with the formation of ion pairs at high proportions of methanol should be the main reason for the change in the size and other characteristics of dendrigrafts in studied mixed water–alcohol solutions. We will check it later when plots for electrostatic properties of the systems at different fractions of methanol will be presented (see Figures in Section 2.2).

Another way to evaluate the size of a dendrigraft is to obtain its hydrodynamic radius. In this work, to estimate the hydrodynamic radius, we used the Kirkwood approximation [102,103]:(2)$$R_{h}^{-1}={\langle r_{ij}^{-1}\rangle }_{i\neq j},$$

It is easy to see that the ratio $R_{h}/R_{g}$ for DGL1 (red) and DGL2 (black) practically does not change in the intervals of ϕ = 0.2–0.6 and increases with an increase in the methanol fraction ϕ≥ 0.7 for DGL2 and at ϕ = 0.9 for DGL1 in agreement with pictures of DGL1 in Figure 1 and DGL2 in Figure 2 as well as with the behavior of $R_{g}$ for both dendrimers in Figure 3a. This ratio for both dendrimers is between two theoretical limits of $R_{h}/R_{g}$ [104]: for the Gaussian coil model (penetrable sphere) with $R_{h}/R_{g}=0.67$ and the rigid sphere model (impenetrable sphere) with $R_{h}/R_{g}=1.29$ shown in Figure 3b by two dashed lines. The effective radius $R_{eff}$ (see Table 1) is an additional characteristic of the size of the dendrigraft (details of its calculation are given below in the section on electrostatic characteristics). The $R_{eff}$ behavior as a function of the methanol volume fraction ϕ is similar to that of the radius of gyration $R_{g}$: it does not change at small ϕ but decreases at ϕ≥ 0.7 for DGL2 and at ϕ = 0.9 for DGL1.

The distribution function $P(R_{g})$ (Figure 4) of the radius of gyration $R_{g}$ demonstrates how often different sizes occur for a given dendrigraft. It is clear that for DGL1 (Figure 4a) at ϕ = 0.2, 0.4, and 0.6, the widths of the distributions and the positions of the maxima are almost the same. For ϕ = 0.7 and 0.8, the maxima are slightly shifted to smaller $R_{g}$ values, and for ϕ = 0.9, there is a significant shift of the maximum for DGL1. These results confirm the results observed in Figure 1 and Figure 3a (for DGL1), where significant changes in $R_{g}$ occur only at ϕ = 0.9. For DGL2 (Figure 4b), the width of the distributions and the position of the maxima are almost the same. However, at ϕ = 0.7, the maximum is shifted to a smaller $R_{g}$ value and for ϕ = 0.8 and 0.9 there are significant shifts of the maxima for DGL2. These results confirm the results observed in Figure 2 and Figure 3a (for DGL2) where changes of size occur already at ϕ = 0.7.

### 2.1. The Local Density

The local structure of the dendrimer around its center of mass (COM) can be estimated using the radial distribution function of the local density
(3)$$\rho \left(r\right)=\frac{1}{4\pi r^{2}}\sum_{i=1}^{N}m_{i}\delta \left(r-r_{i}\right)$$
where δ is the Dirac delta function, *r* is the radial distance from the center of mass of the dendrigraft, $r_{i}$ is the radial distance from the *i*-th atom to the center of mass of the dendrigraft.

Figure 5 shows the radial density distributions for DGL1 at different volume fractions of methanol. We can see a strong change in the shape of the radial density again only at ϕ = 0.9. At the same time, Figure 5b for DGL2 demonstrates a strong change in the shape of the radial density at three fractions of methanol: ϕ = 0.7–0.9. These results are in good agreement with the results of all previous pictures for DGL2, including the distributions $P(R_{g})$ in Figure 4b.

Similar distributions ρ(r) around the dendrigraft center can be calculated for water and methanol molecules. For water molecules, there are flat maximums for DGL1 at *r* = 0.8–1.0 nm (Figure 6a) and DGL2 at 1.2–1.5 nm (Figure 6c). The distributions ρ(r) for methanol molecules for DGL1 (Figure 6b) do not have maxima, but DGL2 (Figure 6d) have maximums at very small *r* = 0.5 nm, close to the center of the dendrigraft at ϕ = 0.2–0.6, and no maximum at higher ϕ. It is interesting that in the case of methanol, there are non-monotonic changes in the density in the regions *r* = 0.5 nm for DGL1 and DGL2 (Figure 6c and Figure 6d, correspondingly) with a change of ϕ. For DGL1, the increase occurs at ϕ = 0.2–0.8 and decreases only at ϕ = 0.9, while for DGL2, an increase is observed in the range ϕ = 0.2–0.6 and decreases at larger ϕ = 0.7–0.9. Thus, significant changes in DGL1 occur only at ϕ = 0.9, while for DGL2, at ϕ = 0.7–0.9. At high *r*, the density curves change monotonically with a change in the volume fraction.

### 2.2. Electrostatic Properties

The analysis of electrostatic characteristics of the system consisting of the dendrigraft and counterion is based on the analysis of the distribution function of the total number of charges in the system (i.e., charges of spherical layers with the radius between *r* and r+dr):(4)$$q\left(r\right)=q_{{\mathrm{NH}}_{3}^{+}}\left(r\right)-q_{{\mathrm{Cl}}^{-}}\left(r\right)$$

The resulting distributions are shown in Figure 7a for DGL1 and Figure 8a for DGL2. Both figures demonstrate the classical structure of the double electrical layer—a characteristic maximum in the positive charge region and a sharp drop to the minimum in the negative charge region. This behavior is explained by the fact that a cloud of negative counterions is formed around a positively charged dendrigraft. It can be seen from Figure 7 that when the fraction of the methanol increases, the height of the maximum and the area under it decrease, and the depth of the minimum and the area above it also become smaller. This occurs due to the additional penetration of counterions into the dendrigraft. As a consequence, the total effective charge of the dendrigraft, together with counterions inside its volume, decreases (due to the increase in the number of counterions in it). The charge outside the dendrigraft (in the negative region) decreases due to the decrease in the number of counterions outside the dendrigraft. We can see from Figure 7a that most strong changes of the charge for DGL1 occur only at ϕ = 0.9, while for DGL2, it occurs at ϕ = 0.7, 0.8, and 0.9. Thus, the strongest change of charge for both dendrigrafts occurs at the same ϕ (ϕ = 0.9 for DGL1 and at ϕ = 0.7, 0.8, and 0.9 for DGL2), where the strongest changes of all conformational properties obtained in previous figures occur. The behaviors of other electrostatic characteristics in Figure 7b–d, as well as in Figure 8b–d, confirm this conclusion (see results below).

We calculated the integral characteristic of the total charge (cumulative charge of spheres of the radius *r* with the center in the center of mass of the dendrigraft).
(5)$$Q\left(r\right)=\int_{0}^{r}q\left(x\right)dx$$

Figure 7b and Figure 8b show radial distributions of cumulative charge Q(r). The value of maximum in this plot corresponds to the uncompensated charge and the position of maximum corresponds to the effective radius $R_{eff}$ of the dendrigraft. The values of $R_{eff}$ were collected in Table 1 for both dendrigrafts at different volume fractions of methanol. The height of the maximum and values of $R_{eff}$ slightly decrease with the increase in the fraction of methanol until ϕ = 0.8 for DGL1 and until ϕ = 0.6 for DGL2. A significant decrease in height of the maximum and $R_{eff}$ occurs for DGL1 only at ϕ = 0.9, while for DGL2 at ϕ = 0.7–0.9. This behavior is very similar to that of $R_{g}$ (Figure 3a).

Solution of the Poisson differential equation
(6)$$\Delta \Psi \left(r\right)=-\frac{4\pi k_{B}T}{e}q\left(r\right),$$
allows us to find the radial distribution of the electrostatic potential. Figure 7c and Figure 8c demonstrates the electrostatic potential around the center of the dendrigraft. It can be seen that electrostatic interactions weaken with an increase in the fraction of methanol at the same fractions of methanol in the mixed solvent as in all previous figures (i.e., at ϕ = 0.9 for DGL1 and at ϕ = 0.7, 0.8, and 0.9 for DGL2).

We also considered the radial distributions of negative charges of counterions $q_{Cl}\left(r\right)$ around dendrigraft center. These distributions are shown in Figure 7d and Figure 8d. It can be seen that when the fraction of methanol increases the maximum of the radial distribution function of the counterion charge for both dendrigrafts shifts to smaller distances *r* relatively center of the dendrigraft. The strongest change of this distribution occurs again at ϕ = 0.9 for DGL1 and at ϕ = 0.7, 0.8, and 0.9 for DGL2. This result confirms results obtained from the analysis of other electrostatic characteristics of both dendrigrafts. Thus, we can conclude that a change in the proportion of methanol in both systems leads to a change in the permittivity of both systems and to a change in their electrical properties, which in turn leads to a change in their conformational properties. The strongest changes of all properties of the dendrigraft occur at high factions of methanol when the effect of the counter ion condensation is strongest.

### 2.3. Hydrogen Bonds

Hydrogen bonding [105,106,107,108] is a specific type of interaction that plays a huge role in biopolymer systems [106,109]. It is well known that these interactions stabilize the secondary structures of proteins and peptides (α-helical and β-sheet conformations), which are the main building blocks of these biological molecules. We considered the distributions of the number of intra-dendrigraft hydrogen bonds as well as the number of hydrogen bonds between the dendrigraft and the solvent.

For both dendrigrafts, an increase in the methanol fraction leads to an increase in the number of inter-dendrigraft hydrogen bonds (Figure 9a for DGL1 and Figure 9c DGL2) and a decrease in the number of hydrogen bonds between the dendrigraft and the solvent (Figure 9b for DGL1 and Figure 9d for DGL2). These results are explained by the well-known fact that each water molecule forms a greater number of hydrogen bonds than a methanol molecule. Therefore, an increase in the volume fraction of methanol in the mixed solvent leads to a decrease in the number of hydrogen bonds between the dendrigraft and the solvent and the possibility of switching the released hydrogen bonds to the formation of intramolecular bonds within the dendrigraft.

### 2.4. Secondary Structure

The secondary structure of linear peptide chain can be specified by pairs of dihedral angles, φ and ψ, describing the rotation around two neighboring chemical bonds of peptide chains connected to C${}_{a}$ carbon [110,111,112].

Ramachandran plots in Figure 10 and Figure 11 demonstrate the presence in the left-half plane (φ < 0) of the two main large areas of secondary structures [113]: (1) “compact α-region” α-helix (ϕ,ψ) = (−60, −40) and other helical structures in this region; (2) “β regions” of elongated structures (consisting of β-sheet (φ,ψ) = (−135, 135) and regions of elongated polyproline II helix (PPII) at (φ,ψ) = (−75, 145). The Ramachandran plots show that for both dendrigrafts the increase in the methanol fraction from 0.2 to 0.6 almost does not change the contribution of the conformation from α-region but leads to the redistribution from the polypropylene II helix to the β-sheet in the “β regions” of extended structures.

A further increase in the methanol fraction from 0.7 to 0.9 leads to an increase in the fraction of compact conformations from the “α-helix region” with ψ from −90 to +90 and a drop in the fraction of elongated structures from “β-region”. To demonstrate this tendency, we made a summation of the number of all points in the Ramachandran plot belonging to two main regions:(1) the region of the compact secondary structures simular to α-helix (φ < 0 and −90 < ψ < 90), and (2) region of the extended secondary structures (β-sheet and the polyproline II helix: (φ < 0 and (90 < ψ and <−90)) at each fraction of methanol in the methanol–water mixture. The resulting plot (Figure 12) demonstrates the dependence of the fraction of compact conformation (“α-helix area”) and extended conformation. It is easy to see that the fraction of the extended conformation is higher and both conformations practically do not change with methanol fractions between ϕ = 0.2 and 0.6. At a methanol fraction of ϕ = 0.7 and higher, the fraction of the compact “α-helix” conformation increases while the fraction of the extended conformation decrease.

## 3. Materials and Methods

In this paper, model dendrigrafts of the first/second generations were considered. The first generation dendrigraft (DGL1) consists of eight linearly linked L-lysine monomers. The symmetrized model of the second-generation dendrigraft (DGL2) includes the same backbone as in DGL1 and eight lysine side chains of five lysines each. Thus, DGL2 consists of 48 lysine residues and its mass and charge are six times greater than those of DGL1 (see Table 2).

Dendrigrafts were simulated in a water–methanol mixture at different volume fractions of methanol. It is well known that methanol contains the same OH group as water and the hydrophobic CH3 group, which makes methanol molecules amphiphilic. Water and methanol also have different compressibility $\kappa_{T}$ [114,115,116,117,118]. Based on these data, we calculate the compressibility as a linear combination of both components of mixed solvent with weights coefficient of components equal to their volume fractions (see Table 3).

For all simulations of DGL1 and DHGL2, in this paper, the AMBER-99SB-ildn force field [119] and the GROMACS package [120] were used. The procedure of the preparation of initial conformations of DGL1 and DGL2 in the methanol–water mixture and the procedure described in our previous simulations was applied [40,41]. The simulated systems consist of the following components: dendrigraft (DGL1 or DGL2), a mixed solvent with a methanol fraction ϕ=0.2,0.4,0.6,0.7,0.8,0.9, and 8 (for DL1) or 48 (for DGL2) Cl counterions. After 250 ns equilibration (see Appendix A), the productive 250 ns MD simulation run was carried out with an integration step 2 fs to obtain the trajectory of the evolution of the systems. The NPT ensemble at P=1 atm, T=310 K, and the mixed solvent compressibility $\kappa_{T}$ (see Table 3) were used in all simulations. The coordinates of atoms were saved every 100 fs for further processing.

We analyzed the simulation results using programs and procedures that were described in our previous papers on the simulation of linear polymers, polyelectrolytes, peptides, polymer brushes, dendrimers, and hyperbranched polymers [121,122,123,124,125,126,127,128,129,130].

## 4. Conclusions

In this work, dendrigrafts of the first and second generations in water–methanol mixtures were studied using the molecular dynamics method. Different volume fractions of methanol were considered. Both dendrigrafts retained their spherical shapes at all methanol fractions. Their sizes and conformational microstructures almost did not change with the increase in the methanol fraction from 0.2 to 0.6. However, a further increase in the methanol fraction from 0.7 to 0.9 led to a decrease in the dendrigraft size and a transition from the extended secondary structures (polyproline II and β-sheet) to an compact α-helix secondary structures. The collapse of the dendrigraft at a high fraction of methanol led to an increase in the internal density and the formation of a greater number of intramolecular hydrogen bonds inside the dendrigrafts. The results could be explained by a smaller dielectric constant of methanol compared with water. Due to this reason, the increase in the volume fraction of methanol led to a decrease in the dielectric constant of the mixed solvent. This led to an increase in the Bjerrum length of the mixed solvent, which is inversely proportional to the dielectric permittivity of the system. The Bjerrum length determines when the counterion condensation occurs, so when it approaches this limit at high methanol fractions the significant changes in all electrical and conformational properties of the dendrigrafts occurs. These results are important for the use of dendrigrafts as drug and gene delivery carriers and the future construction of amphiphilic linear–dendrigraft and dendrigraft–dendrigraft block copolymer systems for drug and gene delivery [47,131,132,133].

## Figures and Tables

**Figure 1 ijms-24-03063-f001:**
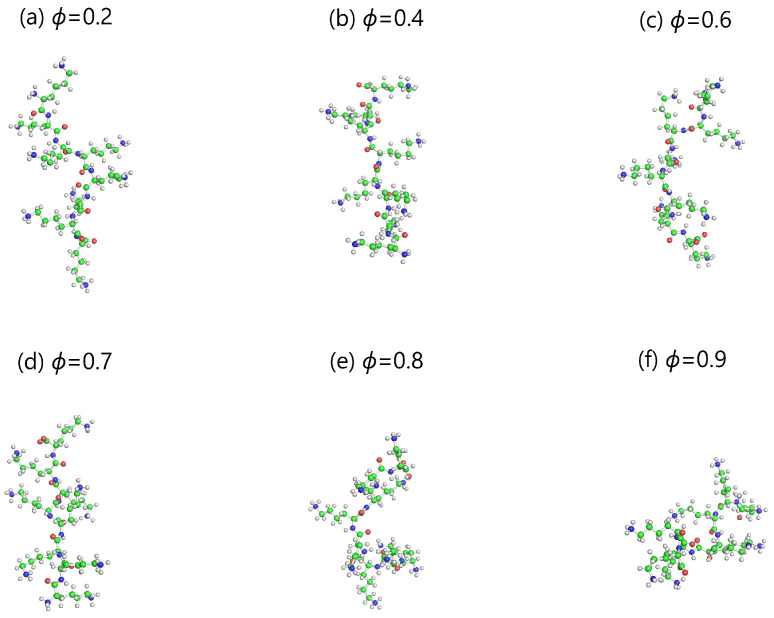
The snapshots of DGL1 at different methanol fractions ϕ. Upper pictures from left to right: 0.2,0.4 and 0.6; bottom pictures from left to right: ϕ=0.7,0.8,0.9.

**Figure 2 ijms-24-03063-f002:**
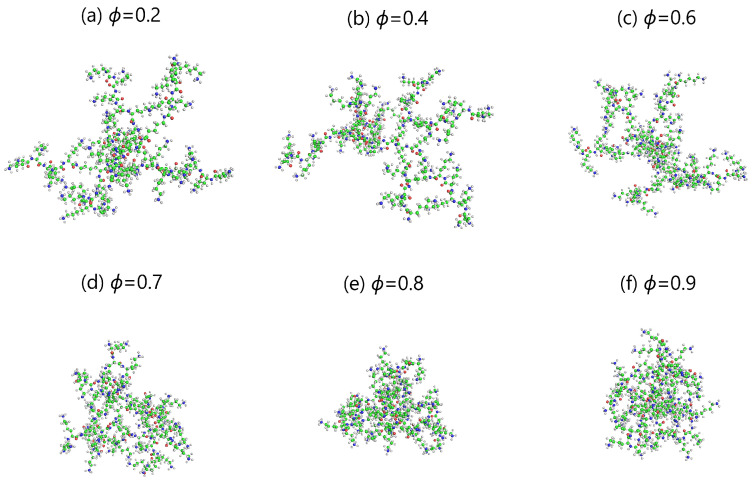
The snapshots of DGL2 at different methanol fractions ϕ. Upper pictures from left to right: 0.2,0.4 and 0.6; bottom pictures from left to right: ϕ=0.7,0.8,0.9.

**Figure 3 ijms-24-03063-f003:**
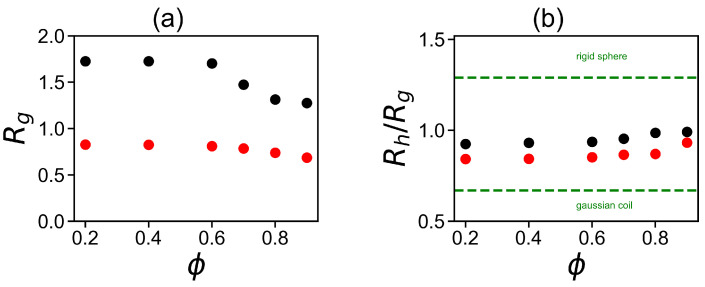
(**a**) The mean-squared radius of gyration $R_{g}$ for DGL1 (red points) and DGL2 (black points); and (**b**) the characteristic ratio of the hydrodynamic radius $R_{h}$ in the Kirkwood approximation to the gyration radius $R_{h}/R_{g}$ for DGL1 (red) and DGL2 (black) as the function of the volume fraction ϕ. Two theoretical limits (the Gaussian coil and the rigid sphere) are depicted in the figure by the dashed lines.

**Figure 4 ijms-24-03063-f004:**
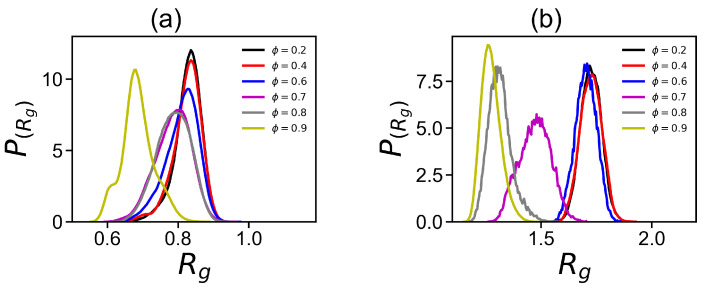
The distribution function $P(R_{g})$ of the radius of gyration $R_{g}$ for (**a**) DGL1 and (**b**) DGL2 at different volume fractions of methanol ϕ in the methanol–water mixture.

**Figure 5 ijms-24-03063-f005:**
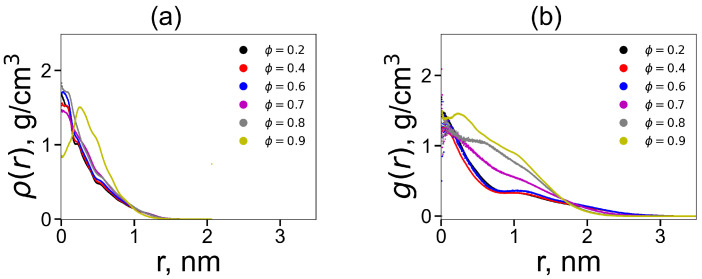
The radial distribution function of density ρ(r) as a function of distance *r* from the center of mass of the dendrigraft at different methanol fractions ϕ: (**a**) DGL1, (**b**) DGL2.

**Figure 6 ijms-24-03063-f006:**
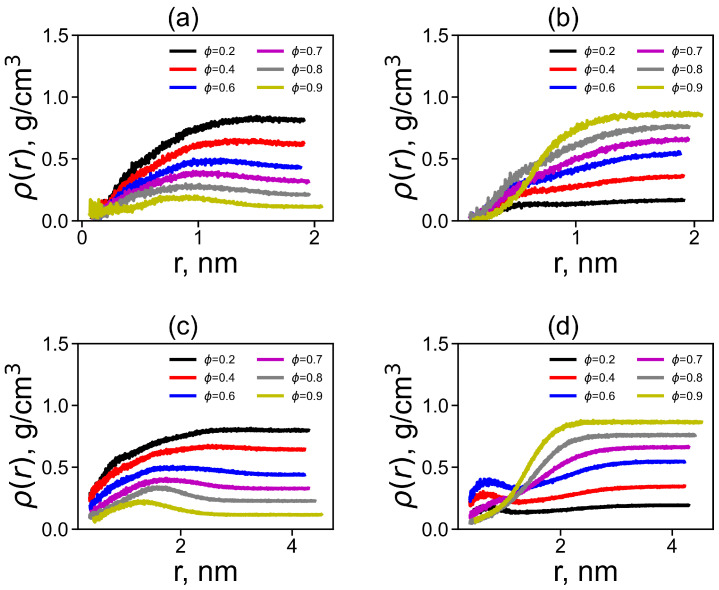
The radial distribution function of the density ρ(r) as a function of distance *r* from the center of mass of the dendrigraft at different methanol fractions for water molecules (**a**) around DGL1, (**c**) around DGL2, for methanol molecules (**b**) around DGL1, and (**d**) around DGL2 at different methanol fractions ϕ.

**Figure 7 ijms-24-03063-f007:**
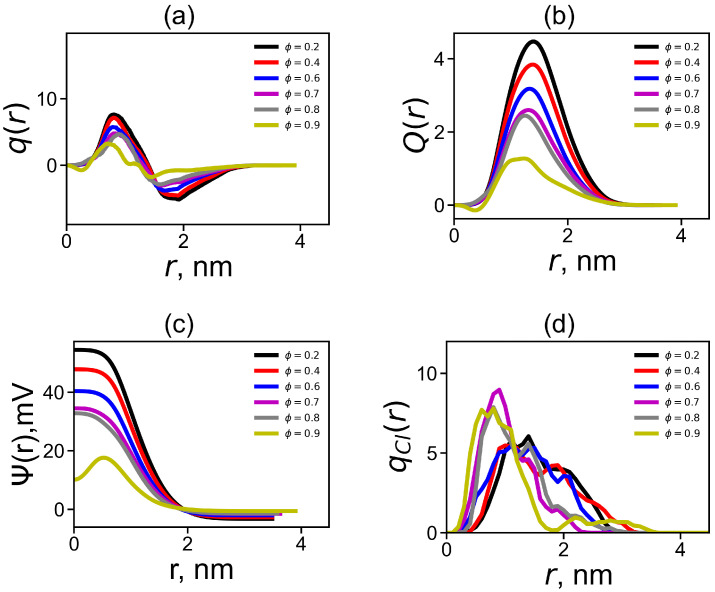
The radial distribution for DGL1 of (**a**) the total charge q(r), (**b**) the cumulative charge Q(r), (**c**) the electrostatic potential Ψ(r), and (**d**) the distribution $q_{Cl}\left(r\right)$ of the counterion number.

**Figure 8 ijms-24-03063-f008:**
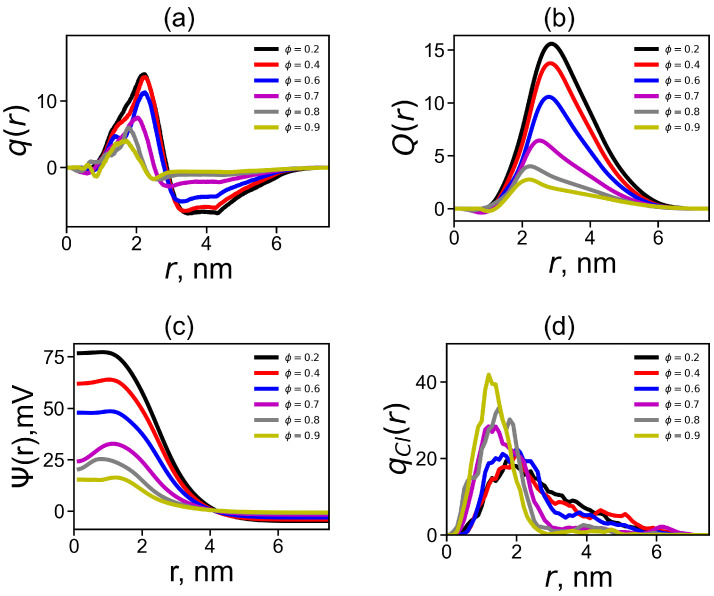
The radial distribution for DGL2 of (**a**) the total charge q(r), (**b**) the cumulative charge Q(r), (**c**) the electrostatic potential Ψ(r), and (**d**) the distribution $q_{Cl}\left(r\right)$ of the counterion number.

**Figure 9 ijms-24-03063-f009:**
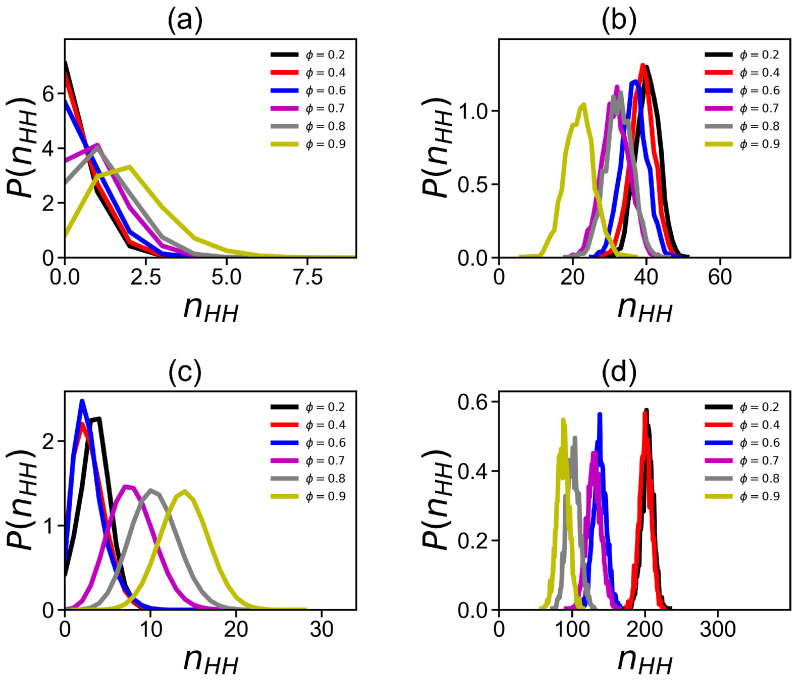
The distribution function of the number of (**a**) intra-dendrigraft hydrogen bonds and (**b**) hydrogen bonds between the dendrigraft and solvent at different methanol fractions ϕ for DGL1. The distribution function of the number of (**c**) intra-dendrigraft hydrogen bonds and (**d**) hydrogen bonds between the dendrigraft and solvent at different methanol fractions ϕ for DGL2.

**Figure 10 ijms-24-03063-f010:**
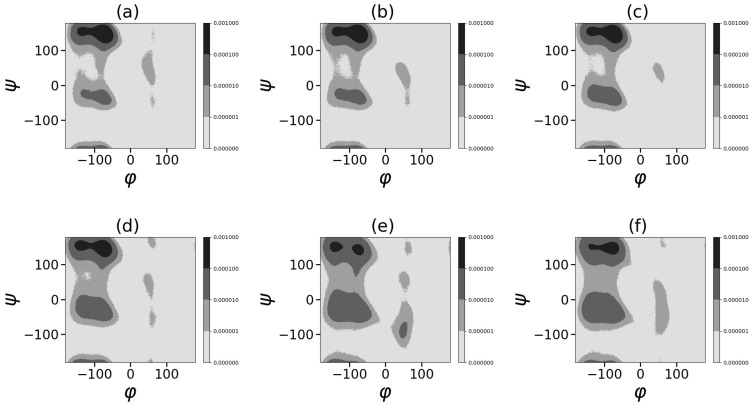
Ramachandran plots. DGL1 at different methanol volume fractions ϕ: (**a**) 0.2, (**b**) 0.4, (**c**) 0.6, (**d**) 0.7, (**e**) 0.8, (**f**) 0.9.

**Figure 11 ijms-24-03063-f011:**
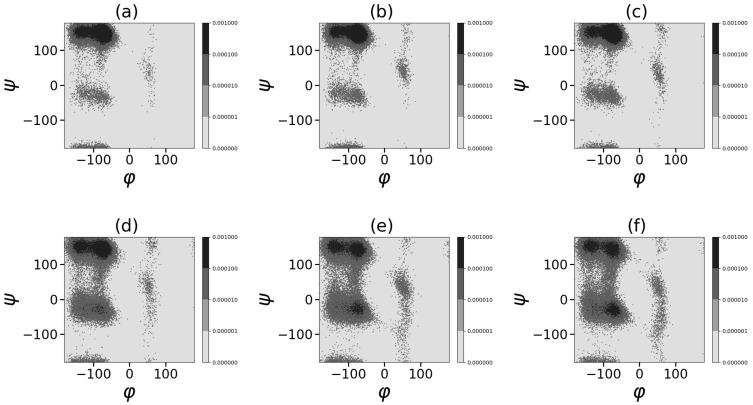
Ramachandran plots. DGL2 at different methanol volume fractions ϕ: (**a**) 0.2, (**b**) 0.4, (**c**) 0.6, (**d**) 0.7, (**e**) 0.8, (**f**) 0.9.

**Figure 12 ijms-24-03063-f012:**
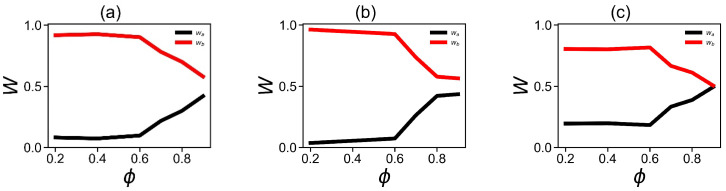
The dependencies of the fraction $w_{a}$ (black line) of all compact secondary structures (α-helix-like) and the $w_{b}$ fraction (red line) of all extended (β-helix-like) conformations versus the methanol fraction ϕ. (**a**) Fractions of corresponding secondary structures in the DGL1 backbone; (**b**) the fraction of corresponding structures in DGL2 side chains; (**c**) the fraction of corresponding structures in the DGL2 backbone.

**Table 1 ijms-24-03063-t001:** The global characteristics of the DGL1 and DGL2 dendrigrafts averaged through the whole trajectory time: the radius of gyration $R_{g}$ (nm), the ratio $R_{h}/R_{g}$, the effective radius $R_{max}$ (nm) (see Section 2.2), and asphericity parameter α at different methanol fractions ϕ.

	DGL1				DGL2			
ϕ	$R_{g}$	$R_{h}/R_{g}$	$R_{\max}$	α	$R_{g}$	$R_{h}/R_{g}$	$R_{\max}$	α
0.2	0.83	0.84	1.39	0.041	1.73	0.92	2.86	0.005
0.4	0.82	0.84	1.38	0.041	1.73	0.93	2.83	0.005
0.6	0.81	0.85	1.33	0.038	1.70	0.94	2.79	0.004
0.7	0.76	0.86	1.31	0.034	1.47	0.95	2.51	0.005
0.8	0.74	0.87	1.24	0.026	1.31	0.98	2.23	0.005
0.9	0.69	0.93	1.23	0.021	1.28	0.99	2.20	0.006

**Table 2 ijms-24-03063-t002:** The characteristics: the molecular mass *M* and the bare charge $Q_{bare}$ of dendrigrafts DGL1 and DGL2, one molecule of water and methanol (MeOH).

	DGL1	DGL2	Water	MeOH
*M* (g/mol)	1051.5	6218.8	18	32
$Q_{bare}$ (e)	+8	+48	0	0

**Table 3 ijms-24-03063-t003:** The characteristics of mixed solvents (water plus methanol): the solvent compressibility $\kappa_{T}$ [1 × 10${}^{-5}$ bar${}^{-1}$] and Bjerrum length $\lambda_{B}$ [nm] at different methanol fractions ϕ.

ϕ	$\kappa_{T}$ [1 × 10${}^{-5}$ bar${}^{-1}$]	$\lambda_{B}$ [nm]
0.2	6.2	0.91
0.4	8.1	1.12
0.6	9.9	1.33
0.7	10.8	1.435
0.8	11.8	1.54
0.9	12.7	1.645

## Data Availability

Not applicable.

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
