# Peer review of "Molecular Dynamics of Lysine Dendrigrafts in Methanol–Water Mixtures"

_ijms, 2023, doi:10.3390/ijms24043063_

Round 1

Reviewer 1 Report

The manuscript described  structure and properties of dendrigrafts of the 1st and 2nd generations in methanol-water mixtures with various volume fractions of methanol. I have a few comments and questions: 

• (page 3) "After equilibration the productive 250 ns MD simulation", what's the sign of equilibration ?

• Suggest to label Ï• value not only in the caption, but also in the  Figure 2 and 3.

• Can the author add more explanation of Ramachandran plots? It seems that in Figure 10 there are clear demarcation in the plots and the two regions are connected, but not in Figure 11  (the two regions are nearly separated). Is that cause by the difference between DGL1 and DGL2? 

• In Figure 9 (a), when Ï• = 0.2, 0.4 and 0.6, the P (nHH) value decreases, but when Ï• = 0.7, 0.8 and 0.9 the P (nHH) value increases and then decreases. Is there any explanation of this phenomenon?

• DGL2 has side chains, but DGL1 doesn't. The author described the fraction of residues in DGL2 side chains in Figure 12. Is there any other calculation can be performed to study the impact of side chain on the property of dendrigrafts?

Reviewer 2 Report

"Molecular dynamics of lysine dendrigrafts in methanol-water mixtures" by Fatullaev et al studied the influence of methanol ratio on the properties of lysine dendrigrafts, including size, local structure, electrostatic properties, hydrogen bonds, etc. in methanol-water mixtures using molecular dyanmics simulation. The motivation of this manuscript needs to be further elaborated and the results need further in-depth analysis. The authors should address the following issues listed below.

1. The author stated in the introduction that "There are very few publications in the literature devoted to modeling dendrigrafts in water, and there are practically no publications describing their behavior in other solvents and their mixtures with water, including its mixtures with methanol. " Excep "no one has done such research" as the authors suggested, could the authors explain why they want to study the lysine dendrigrafts in the water/methanol mixed solvents? Are there any pratical application of lysine dendrigrafts in the mixed solvents? Did the authors simulate other solvent molecules except methanol?

2. The authors tested different volume ratios of methanol in the water/methanol mixed solvents. It can be seen from the results that the methanol fraction affects many properties of lysine dendrigrafts. Could the authors suggest which ratio(s) are optimal in different kinds of potential applications?

3. Pages 4-5, the authors used gyration radius and ratio Rh/Rg to study the dendrigraft size. Could the authors explain why analyzing the dendrigraft size? Does the size have any influences on any properties? Furthermore, the authors only showed the size differences at different methanol ratios and didn't provide any further analysis. It would be helpful to provide in-depth analysis on why different methanol ratios result in different dendrigraft sizes.

4. Section "The local density". The authors should state why studying the local density and why it's important. Also the authors should illustrate why methanol ratios cause the change in the shape of radial density, rather than just displying the results. Similarly, in section "Electrostatic properties", the authors should state why studying it and how methanol ratios affect the electrostatic properties.

5. Line 126-128, "Figure 6" should be Figure 5? 

6. Line 155, "Therefore, an increase of fraction of methanol in methanol-water mixture leads to a decrease in the dielectric constant of the mixed solvent, an increase in the Bjerrum length in it, and to the condensation of negative counterions on the positive charges of the dendrigraft." This sentence needs rephrasing for clarity.

7. Line 171, "It can be seen that when fraction of methanol increase the maximum of radial distribution function of counterion charge for both dendrigrafts shifts to smaller distances r relatively center of dendrigraft." Could the authors come up with an assumption why is this?

8. Line 177, "Hydrogen bonding [122–125] is a specific type of interaction that plays a huge role in biopolymer systems." It would be more clear to the readers why hydrogen bond is studied here if the authors listed out what "huge roles" does the hydrogen bond play.

Round 2

Reviewer 2 Report

The revised material is now more scientifically sound. The following ideas from the authors' cover letter could also be included in the paper.

1. Response #2 "it is best to use the range of methanol volume fractions from 0.2 to 0.6, and the optimal value is a fraction equal to 0.4."

2. Response #4. The analysis of local density could be added to the manuscript.